# Learning Sensor Multiplexing Design through Back-propagation

**Ayan Chakrabarti**
Toyota Technological Institute at Chicago
6045 S. Kenwood Ave., Chicago, IL
ayanc@ttic.edu

## Abstract

Recent progress on many imaging and vision tasks has been driven by the use of deep feed-forward neural networks, which are trained by propagating gradients of a loss defined on the final output, back through the network up to the first layer that operates directly on the image. We propose back-propagating one step further—to learn camera sensor designs jointly with networks that carry out inference on the images they capture. In this paper, we specifically consider the design and inference problems in a typical color camera—where the sensor is able to measure only one color channel at each pixel location, and computational inference is required to reconstruct a full color image. We learn the camera sensor's color multiplexing pattern by encoding it as layer whose learnable weights determine which color channel, from among a fixed set, will be measured at each location. These weights are jointly trained with those of a reconstruction network that operates on the corresponding sensor measurements to produce a full color image. Our network achieves significant improvements in accuracy over the traditional Bayer pattern used in most color cameras. It automatically learns to employ a sparse color measurement approach similar to that of a recent design, and moreover, improves upon that design by learning an optimal layout for these measurements.

## 1 Introduction

With the availability of cheap computing power, modern cameras can rely on computational post-processing to extend their capabilities under the physical constraints of existing sensor technology. Sophisticated techniques, such as those for denoising [3, 28], deblurring [19, 26], etc., are increasingly being used to improve the quality of images and videos that were degraded during acquisition. Moreover, researchers have posited novel sensing strategies that, when combined with post-processing algorithms, are able to produce higher quality and more informative images and videos. For example, coded exposure imaging [18] allows better inversion of motion blur, coded apertures [14, 23] allow passive measurement of scene depth from a single shot, and compressive measurement strategies [1, 8, 25] combined with sparse reconstruction algorithms allow the recovery of visual measurements with higher spatial, spectral, and temporal resolutions.

Key to the success of these latter approaches is the co-design of sensing strategies and inference algorithms, where the measurements are designed to provide information complimentary to the known statistical structure of natural scenes. So far, sensor design in this regime has largely been either informed by expert intuition (*e.g.*, [4]), or based on the decision to use a specific image model or inference strategy—*e.g.*, measurements corresponding to random [1], or dictionary-specific [5], projections are a common choice for sparsity-based reconstruction methods. In this paper, we seek to enable a broader data-driven exploration of the joint sensor and inference method space, by learning both sensor design and the computational inference engine end-to-end.

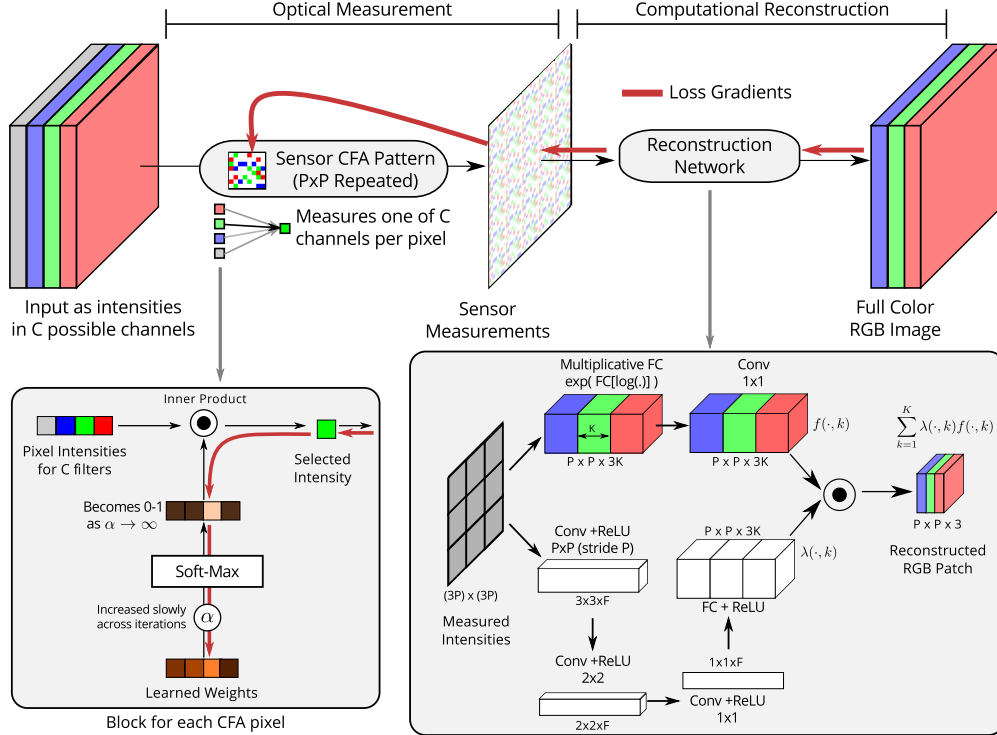

Figure 1: We propose a method to learn the optimal color multiplexing pattern for a camera through joint training with a neural network for reconstruction. (**Top**) Given $C$ possible color filters that could be placed at each pixel, we parameterize the incident light as a $C-$channel image. This acts as input to a "sensor layer" that learns to select one of these channel at each pixel. A reconstruction network then processes these measurements to yield a full-color RGB image. We jointly train both for optimal reconstruction quality. (**Bottom left**) Since the hard selection of individual color channels is not differentiable, we encode these decisions using a Soft-max layer, with a "temperature" parameter $\alpha$ that is increased across iterations. (**Bottom right**) We use a bifurcated architecture with two paths for the reconstruction network. One path produces $K$ possible values for each color intensity through multiplicative and linear interpolation, and the other weights to combine these into a single estimate.

We leverage the successful use of back-propagation and stochastic gradient descent (SGD) [13] in learning deep neural networks for various tasks [12, 16, 20, 24]. These networks process a given input through a complex cascade of layers, and training is able to jointly optimize the parameters of all layers to enable the network to succeed at the final inference task. Treating optical measurement and computational inference as a cascade, we propose using the same approach to learn both jointly. We encode the sensor's design choices into the learnable parameters of a "sensor layer" which, once trained, can be instantiated by camera optics. This layer's output is fed to a neural network that carries out inference computationally on the corresponding measurements. Both are then trained jointly.

We demonstrate this approach by applying it to the sensor-inference design problem in a standard digital color camera. Since image sensors can physically measure only one color channel at each pixel, cameras spatially multiplex the measurement of different colors across the sensor plane, and then computationally recover the missing intensities through a reconstruction process known as demosaicking. We jointly learn the spatial pattern for multiplexing different color channels—that requires making a hard decision to use one of a discrete set of color filters at each pixel—along with a neural network that performs demosaicking. Together, these enable the recovery of high-quality color images of natural scenes. We find that our approach significantly outperforms the traditional Bayer pattern [2] used in most color cameras. We also compare it to a recently introduced design [4] based on making sparse color measurements, that has superior noise performance and fewer aliasing artifacts. Interestingly, our network automatically learns to employ a similar measurement strategy, but is able outperform this design by finding a more optimal spatial layout for the color measurements.

## 2 Background

Since both CMOS and CCD sensors can measure only the total intensity of visible light incident on them, color is typically measured by placing an array of color filters (CFA) in front of the sensor plane. The CFA pattern determines which color channel is measured at which pixel, with the most commonly pattern used in RGB color cameras being the Bayer mosaic [2] introduced in 1976. This is a $4 \times 4$ repeating pattern, with two measurements of the green channel and one each of red and blue. The color values that are not directly measured are then reconstructed computationally by demosaciking algorithms. These algorithms [15] typically rely on the assumption that different color channels are correlated and piecewise smooth, and reason about locations of edges and other high-frequency image content to avoid creating aliasing artifacts.

This approach yields reasonable results, and the Bayer pattern remains in widespread use even today. However, the choice of the CFA pattern involves a trade-off. Color filters placed in front of the sensor block part of the incident light energy, leading to longer exposure times or noisier measurements (in comparison to grayscale cameras). Moreover, since every channel is regularly sub-sampled in the Bayer pattern, reconstructions are prone to visually disturbing aliasing artifacts even with the best reconstruction methods. Most consumer cameras address this by placing an anti-aliasing filter in front of the sensor to blur the incident light field, but this leads to a loss of sharpness and resolution.

To address this, Chakrabarti *et al.* [4] recently proposed the use of an alternative CFA pattern in which a majority of the pixels measure the total unfiltered visible light intensity. Color is measured only sparsely, using $2 \times 2$ Bayer blocks placed at regularly spaced intervals on the otherwise unfiltered sensor plane. The resulting measured image corresponds to an un-aliased full resolution luminance image (*i.e.*, the unfiltered measurements) with "holes" at the color sampling site; with point-wise color information on a coarser grid. The reconstruction algorithm in [4] is significantly different from traditional demosaicking, and involves first recovering missing luminance values by hole-filling (which is computationally easier than up-sampling since there is more context around the missing intensities), and then propagating chromaticities from the color measurement sites to the remaining pixels using edges in the luminance image as a guide. This approach was shown to significantly improve upon the capabilities of a Bayer sensor—in terms of better noise performance, increased sharpness, and reduced aliasing artifacts.

That [4]'s CFA pattern required a very different reconstruction algorithm illustrates the fact that both the sensor and inference method need to be modified together to achieve gains in performance. In [4]'s case, this was achieved by applying an intuitive design principles—of making high SNR non-aliased measurements of one color channel. However, these principles are tied to a specific reconstruction approach, and do not tell us, for example, whether regularly spaced $2 \times 2$ blocks are the optimal way of measuring color sparsely.

While learning-based methods have been proposed for demosaicking [10, 17, 22] (as well as for joint demosaicking and denoising [9, 11]), these work with a pre-determined CFA pattern and training is used only to tune the reconstruction algorithm. In contrast, our approach seeks to learn, automatically from data, *both* the CFA pattern and reconstruction method, so that they are jointly optimal in terms of reconstruction quality.

## 3 Jointly Learning Measurement and Reconstruction

We formulate our task as that of reconstructing an RGB image $y(n) \in \mathbb{R}^3$, where $n \in \mathbb{Z}^2$ indexes pixel location, from a measured sensor image $s(n) \in \mathbb{R}$. Along with this reconstruction task, we also have to choose a multiplexing pattern which determines the color channel that each $s(n)$ corresponds to. We let this choice be between one of $C$ channels—a parameterization that takes into account which spectral filters can be physically synthesized. We use $x(n) \in \mathbb{R}^C$ denote the intensity measurements corresponding to each of these color channels, and a zero-one selection map $I(n) \in \{0,1\}^C, |I(n)| = 1$ to encode the multiplexing pattern, such that the corresponding sensor measurements are given by $s(n) = I(n)^T x(n)$. Moreover, we assume that $I(n)$ repeats periodically every $P$ pixels, and therefore only has $P^2$ unique values.

Given a training set consisting of pairs of output images $y(n)$ and $C$-channel input images $x(n)$, our goal then is to learn this pattern $I(n)$, jointly with a reconstruction algorithm that maps the corresponding measurements $s(n)$ to the full color image output $y(n)$. We use a neural network

to map sensor measurements $s(n)$ to an estimate $\hat{y}(n)$ of the full color image. Furthermore, we encode the measurement process into a "sensor layer", which maps the input $x(n)$ to measurements $s(n)$, and whose learnable parameters encode the multiplexing pattern $I(n)$. We then learn both the reconstruction network and the sensor layer simultaneously, with respect to a squared loss $\|\hat{y}(n) - y(n)\|^2$ between the reconstructed and true color images.

## 3.1 Learning the Multiplexing Pattern

The key challenge to our joint learning problem lies in recovering the optimal multiplexing pattern $I(n)$, since it is ordinal-valued and requires learning to make a hard non-differentiable decision between $C$ possibilities. To address this, we rely on the standard soft-max operation, which is traditionally used in multi-label classification tasks.

However, we are unable to use the soft-max operation directly—unlike in classification tasks where the ordinal labels are the final output, and where the training objective prefers hard assignment to a single label, in our formulation $I(n)$ is used to generate sensor measurements that are then processed by a reconstruction network. Indeed, when using a straight soft-max, we find that the reconstruction network converges to real-valued $I(n)$ maps that correspond to measuring different weighted combinations of the input channels. Thresholding the learned $I(n)$ to be ordinal valued leads to a significant drop in performance, even when we further train the reconstruction network to work with this thresholded version.

Our solution to this is fairly simple. We use a soft-max with a temperature parameter that is increased slowly through training iterations. Specifically, we learn a vector $w(n) \in \mathbb{R}^C$ for each location $n$ of the multiplexing pattern, with the corresponding $I(n)$ given during training as:

$$I(n) = \text{Soft-max}\left[\alpha_t w(n)\right], \tag{1}$$

where $\alpha_t$ is a scalar factor that we increase with iteration number $t$.

Therefore, in addition to changes due to the SGD updates to $w(n)$, the effective distribution of $I(n)$ become "peakier" at every iteration because of the increasing $\alpha_t$, and as $\alpha_t \to \infty$, $I(n)$ becomes a zero-one vector. Note that the gradient magnitudes of $w(n)$ also scale-up, since we compute these gradients at each iteration with respect to the current value of $t$. This ensures that the pattern can keep learning in the presence of a strong supervisory signal from the loss, while retaining a bias to drift towards making a hard choice for a single color channel.

As illustrated in Fig. 1, our sensor layer contains a parameter vector $w(n)$ for each pixel of the $P \times P$ multiplexing pattern. During training, we generate the corresponding $I(n)$ vectors using (1) above, and the layer then outputs sensor measurements based on the $C$-channel input $x(n)$ as $s(n) = I(n)^T x(n)$. Once training is complete (and for validation during training), we replace $I(n)$ with its zero-one version as $I(n)^c = 1$ for $c = \arg\max_c w^c(n)$, and 0 otherwise.

As we report in Sec. 4, our approach is able to successfully learn an optimal sensing pattern, which adapts during training to match the evolving reconstruction network. We would also like to note here two alternative strategies that we explored to learn an ordinal $I(n)$, which were not as successful. We considered using a standard soft-max approach with a separate entropy penalty on the distribution $I(n)$—however, this caused the pattern $I(n)$ to stop learning very early during training (or for lower weighting of the penalty, had no effect at all). We also tried to incrementally pin the lowest $I(n)$ values to zero after training for a number of iterations, in a manner similar to Han *et al.*'s [7] approach to network compression. However, even with significant tuning, this approach caused a large parts of the pattern search space to be eliminated early, and was not able to adapt to the fact that a channel with a low weight at a particular location might eventually become desirable based on changes to the pattern at other locations, and corresponding updates to the reconstruction network.

## 3.2 Reconstruction Network Architecture

Traditional demosaicking algorithms [15] produce a full color image by interpolating the missing color values from neighboring measurement sites, and by exploiting cross-channel dependencies. This interpolation is often linear, but in some cases takes the form of transferring chromaticities or color ratios (*e.g.*, in [4]). Moreover, most demosaicking algorithms reason about image textures and edges to avoid smoothing across boundaries or creating aliasing artifacts.

We adopt a simple bifurcated network architecture that leverages these intuitions. As illustrated in Fig. 1, our network reconstructs each $P \times P$ patch in $y(n)$ from a receptive field that is centered on that patch in the measured image $s(n)$, and thrice as large in each dimension. The network has two paths, both of operate on the entire input and both output $(P \times P \times 3K)$ values, *i.e.*, $K$ values for each output color intensity. We denote these outputs as $\lambda(n, k), f(n, k) \in \mathbb{R}^3$.

One path produces $f(n, k)$ by first computing multiplicative combinations of the entire $3P \times 3P$ input patch—we instantiate this using a fully-connected layer without a bias term that operates in the log-domain—followed by a linear combinations across each of the $3K$ values at each location. We interpret these $f(n, k)$ values as $K$ proposals for each $y(n)$. The second path uses a more standard cascade of convolution layers—all of which have $F$ outputs with the first layer having a stride of $P$—followed by a fully connected layer that produces the outputs $\lambda(n, k)$ with the same dimensionality as $f(n, k)$. We treat $\lambda(n, k)$ as gating values for the proposals $f(n, k)$, and generate the final reconstructed patch $\hat{y}(n)$ as $\sum_k \lambda(n, k) f(n, k)$.

## 4    Experiments

We follow a similar approach to [4] for training and evaluating our method. Like [4], we use the Gehler-Shi database [6, 21] that consists of 568 color images of indoor and outdoor scenes, captured under various illuminants. These images were obtained from RAW sensor images from a camera employing the Bayer pattern with an anti-aliasing optical filter, by using the different color measurements in each Bayer block to construct a single RGB pixel. These images are therefore at half the resolution of the original sensor image, but have statistics that are representative of aliasing-free full color images of typical natural scenes. Unlike [4] who only used 10 images for evaluation, we use the entire dataset—using 56 images for testing, 461 images for training, and the remaining 51 images as a validation set to fix hyper-parameters.

We treat the images in the dataset as the ground truth for the output RGB images $y(n)$. As sensor measurements, we consider $C = 4$ possible color channels. The first three correspond to the original sensor RGB channels. Like [4], we choose the fourth channel to be white or panchromatic, and construct it as the sum of the RGB measurements. As mentioned in [4], this corresponds to a conservative estimate of the light-efficiency of an unfiltered channel. We construct the $C$-channel input image $x(n)$ by including these measurements, followed by addition of different levels of Gaussian noise, with high noise variances simulating low-light capture.

We learn a repeating pattern with $P = 8$. In our reconstruction network, we set the number of proposals $K$ for each output intensity to 24, and the number of convolutional layer outputs $F$ in the second path of our network to 128. When learning our sensor multiplexing pattern, we increase the scalar soft-max factor $\alpha_t$ in (1) according to a quadratic schedule as $\alpha_t = 1 + (\gamma t)^2$, where $\gamma = 2.5 \times 10^{-5}$ in our experiments. We train a separate reconstruction network for each noise level (positing that a camera could select between these based on the ISO settings). However, since it is impractical to employ different sensors for different settings, we learn a single spatial multiplexing pattern, optimized for reconstruction under moderate noise levels with standard deviation (STD) of 0.01 (with respect to intensity values in $x(n)$ scaled to be between 0 and 1).

We train our sensor layer and reconstruction network jointly at this noise level on sets of $8 \times 8$ $y(n)$ patches and corresponding $24 \times 24$ $x(n)$ patches sampled randomly from the training set. We use a batch-size of 128, with a learning rate of 0.001 for 1.5 million iterations. Then, keeping the sensor pattern fixed to our learned version, we train reconstruction networks from scratch for other noise levels—training again with a learning rate of 0.001 for 1.5 million iterations, followed another 100,000 iterations with a rate of $10^{-4}$. We also train reconstruction networks at all noise levels in a similar way for the Bayer pattern, as well the pattern of [4] (with a color sampling rate of 4). Moreover, to allow consistent comparisons, we re-train the reconstruction network for our pattern at the 0.01 noise level from scratch following this regime.

### 4.1    Evaluating the Reconstruction Network

We begin by comparing the performance of our learned reconstruction networks to traditional demosaicking algorithms for the standard Bayer pattern, and the pattern of [4]. Note that our goal is not to propose a new demosaicking method for existing sensors. Nevertheless, since our sensor

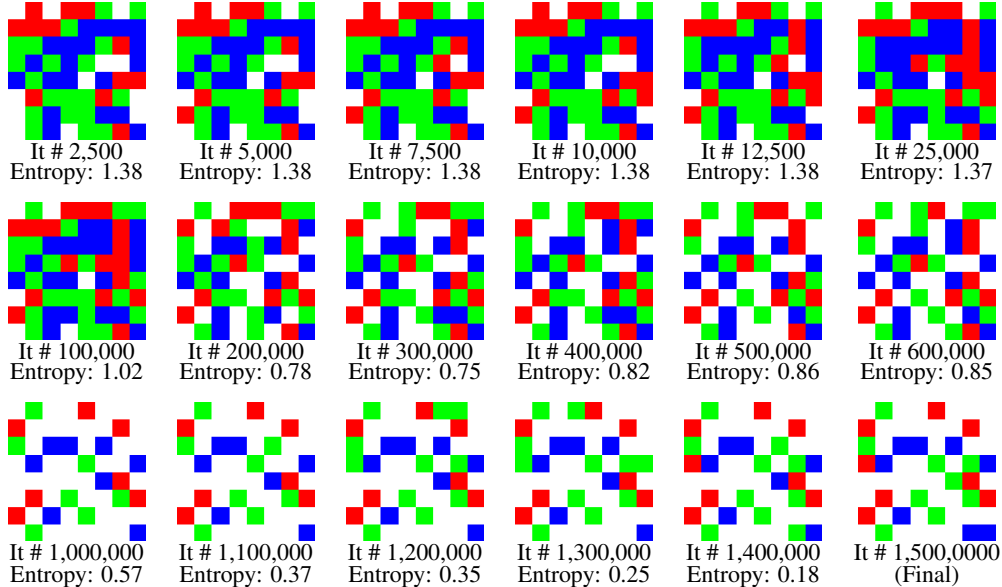

Figure 2: Evolution of sensor pattern through training iterations. We find that the our network's color sensing pattern changes qualitatively through the training process. In initial iterations, the sensor layer learns to sample color channels directly. As training continues, these color measurements are replaced by panchromatic (white) pixels. The final iterations see fine refinements to the pattern. We also report the mean (across pixels) entropy of the underlying distribution $I(n)$ for each pattern. Note that, as expected, this entropy decreases across iterations as the distributions $I(n)$ evolve from being soft selections of color channels, to zero-one vectors that make hard ordinal decisions.

Table 1: Median Reconstruction PSNR (dB) using Traditional demosaicking and Proposed Network

|  | Bayer | | CFZ [4] | |
| --- | --- | --- | --- | --- |
|  | Noise STD=0.0025 | Noise STD=0.01 | Noise STD=0.0025 | Noise STD=0.01 |
| Traditional | 42.69 | 32.44 | 48.84 | 39.55 |
| Network | 47.55 | 43.72 | 49.08 | 44.64 |

pattern is being learned jointly with our proposed reconstruction architecture, it is important to determine whether this architecture can learn to reason effectively with different kinds of sensor patterns, which is necessary to effectively cover the joint sensor-inference design space.

We compare our learned networks to Zhang and Wu's method [27] for the Bayer pattern, and Chakrabarti *et al.*'s method [4] for their own pattern. We measure performance in terms of the reconstruction PSNR of all non-overlapping $64 \times 64$ patches from all test images (roughly 40,000 patches). Table 1 compares the median PSNR values across all patches for reconstructions using our network to those from traditional methods, at two noise levels—low noise corresponding to an STD of 0.0025, and moderate noise corresponding to 0.01. For the pattern of [4], we find that our network performs similar to their reconstruction method at the low noise level, and significantly better at the higher noise level. On the Bayer pattern, our network achieves much better performance at both noise levels. We also note here that reconstruction using our network is significantly faster—taking 9s on a six core CPU, and 200ms when using a Titan X GPU, for a 2.7 mega-pixel image. In comparison, [4] and [27]'s reconstruction methods take 20s and 1 min. respectively on the CPU.

## 4.2 Visualizing Sensor Pattern Training

In Fig. 2, we visualize the evolution of our sensor pattern during the training process, while it is being jointly learned with the reconstruction network. In the initial iterations, the sensor layers displays a preference for densely sampling the RGB channels, with very few panchromatic measurements—in fact, in the first row of Fig. 2, we see panchromatic pixels switching to color measurements. This

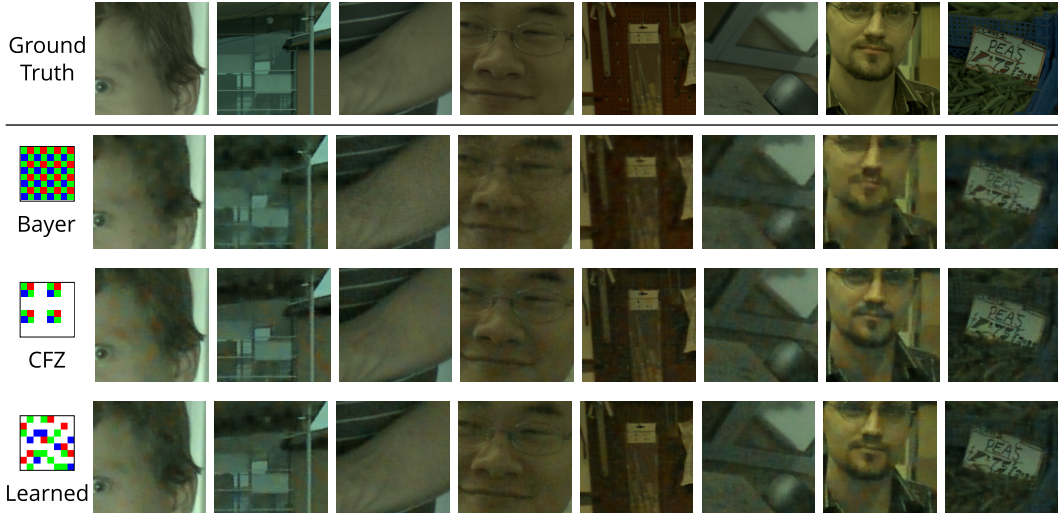

Figure 3: Example reconstructions from (noisy) measurements with different sensor multiplexing patterns. Best viewed at higher resolution in the electronic version.

is likely because early on in the training process, the reconstruction network hasn't yet learned to exploit cross-channel correlations, and therefore needs to measure the output channels directly.

However, as training progresses, the reconstruction network gets more sophisticated, and we see the number of color measurements get sparser and sparser, in favor of panchromatic pixels that offer the advantage of higher SNR. Essentially, the sensor layer begins to adopt one of the design principles of [4]. However, it distributes the color measurement sites across the pattern, instead of concentrating them into separated blocks like [4]. In the last 500K iterations, we see that most changes correspond to fine refinements of the pattern, with a few individual pixels swapping the channels they measure.

While the patterns themselves in Fig. 2 correspond to the channel at each pixel with the maximum value in the selection map $I(n)$, remember that these maps themselves are soft. Therefore, we also report the mean entropy of the underlying $I(n)$ for each pattern in Fig. 2. We see that this entropy decreases across iterations, as the choice of color channel for more and more pixels becomes fixed, with their distributions in $I(n)$ becoming peakier and closer to being zero-one vectors.

### 4.3 Evaluating Learned Pattern

Finally, we evaluate the performance of neural network-based reconstruction from measurements with our learned pattern, to those with the Bayer pattern and the pattern of [4]. Table 2 shows different quantiles of reconstruction PSNR for various noise levels, with noise STDs raning from 0 to 0.04. Even though our sensor pattern was trained at the noise level of STD=0.01, we find it achieves the highest reconstruction quality over a large range of noise levels. Specifically, it always outperforms the Bayer pattern, by fairly significant margins at higher noise levels. The improvement in performance over [4]'s pattern is less pronounced, although we do achieve consistently higher PSNR values for all quantiles at most noise levels. Figure 3 shows examples of color patches reconstructed from our learned sensor, and compare these to those from the Bayer pattern and [4].

We see that the reconstructions from the Bayer pattern are noticeably worse. This is because it makes lower SNR measurements, and the reconstruction networks learn to smooth their outputs to reduce this noise. Both [4] and our pattern yield significantly better reconstructions. Indeed, most of our gains over the Bayer pattern come from choosing to make most measurements panchromatic, a design principle shared by [4]. However, remember that our sensor layer learns this principle entirely automatically from data, without expert supervision. Moreover, we see that [4]'s reconstructions tend to have a few more instances of "chromaticity noise", in the form of contiguous regions with incorrect hues, which explain its slightly lower PSNR values in Table 2.

Table 2: Network Reconstruction PSNR (dB) Quantiles for various CFA Patterns

| Noise STD | Percentile | Bayer [2] | CFZ [4] | Learned |
|---|---|---|---|---|
| 0 | 25% | 47.62 | **48.04** | 47.97 |
| | 50% | 51.72 | **52.17** | 52.12 |
| | 75% | 54.97 | **55.32** | 55.30 |
| 0.0025 | 25% | 44.61 | 46.05 | **46.08** |
| | 50% | 47.55 | 49.08 | **49.17** |
| | 75% | 50.52 | 51.57 | **51.76** |
| 0.0050 | 25% | 42.55 | 44.33 | **44.37** |
| | 50% | 45.63 | 47.01 | **47.19** |
| | 75% | 48.73 | 49.68 | **49.94** |
| 0.0075 | 25% | 41.34 | 42.92 | **43.08** |
| | 50% | 44.48 | 45.60 | **45.85** |
| | 75% | 47.77 | 48.41 | **48.69** |
| 0.0100 | 25% | 40.58 | 41.97 | **42.16** |
| | 50% | 43.72 | 44.64 | **44.94** |
| | 75% | 47.10 | 47.56 | **47.80** |
| 0.0125 | 25% | 40.29 | 41.17 | **41.41** |
| | 50% | 43.36 | 43.88 | **44.22** |
| | 75% | 46.65 | 47.04 | **47.27** |
| 0.0150 | 25% | 39.97 | 40.54 | **40.85** |
| | 50% | 43.03 | 43.29 | **43.69** |
| | 75% | 46.25 | 46.69 | **46.86** |
| 0.0175 | 25% | 39.60 | 40.03 | **40.31** |
| | 50% | 42.62 | 42.83 | **43.12** |
| | 75% | 45.82 | 46.39 | **46.45** |
| 0.0200 | 25% | 39.31 | 39.49 | **39.96** |
| | 50% | 42.39 | 42.39 | **42.78** |
| | 75% | 45.56 | 46.14 | **46.23** |
| 0.0300 | 25% | 38.18 | 38.31 | **38.92** |
| | 50% | 41.17 | 41.48 | **41.85** |
| | 75% | 44.23 | 45.61 | **45.63** |
| 0.0400 | 25% | 37.14 | 37.43 | **38.00** |
| | 50% | 39.98 | 40.86 | **41.02** |
| | 75% | 43.17 | **45.11** | 44.98 |

# 5  Conclusion

In this paper, we proposed learning sensor design jointly with a neural network that carried out inference on the sensor's measurements, specifically focusing on the problem of finding the optimal color multiplexing pattern for a digital color camera. We learned this pattern by joint training with a neural network for reconstructing full color images from the multiplexed measurements. We used a soft-max operation with an increasing temperature parameter to model the non-differentiable color channel selection at each point, which allowed us to train the pattern effectively. Finally, we demonstrated that our learned pattern enabled better reconstructions than past designs. An implementation of our method, along with trained models, data, and results, is available at our project page at http://www.ttic.edu/chakrabarti/learncfa/.

Our results suggest that learning measurement strategies jointly with computational inference is both useful and possible. In particular, our approach can be used directly to learn other forms of optimized multiplexing patterns—*e.g.*, spatio-temporal multiplexing for video, viewpoint multiplexing in light-field cameras, etc. Moreover, these patterns can be learned to be optimal for inference tasks beyond reconstruction. For example, a sensor layer jointly trained with a neural network for classification could be used to discover optimal measurement strategies for say, distinguishing between biological samples using multi-spectral imaging, or detecting targets in remote sensing.

**Acknowledgments**

We thank NVIDIA corporation for the donation of a Titan X GPU used in this research.

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
