[Reviews · NeurIPS 2016]

Reviewer 1

Summary

Learning Sensor Multiplexing Design through Back-propagation This paper proposes to learn jointly i) the color filter pattern of RGB camera sensors and ii) the color reconstruction of a final output image, using a neural network framework. Contributions have potential high impact in camera designing, since the concept of automatically learning the color filter pattern moves the industry beyond the arbitrary Bayer filter, and the jointly learned color reconstruction offers better image quality in terms of psnr. Experiments are following previous literature and evaluate noise distributions in various camera sensor designs.

Qualitative Assessment

Paper is nicely written, methodology is easy to follow, and validation shows the benefit of the proposed approach. Comments below: One main question that could arise in learning automatically a color filter pattern, is the dependence on the dataset. Here benchmarking was performed with the Gehler-Shi dataset. Would the use of a different heterogeneous dataset produce a completely different sensor pattern? If the pattern changes significantly between training sets, would it be relevant to question the robustness of a chosen pattern? On the same note, how could this impact camera designs, ie., could a universal pattern exist, or would there be multiple patterns, each designed for particular situations (could patterns designed for outdoor, indoor, urban or nature settings, produce a much better image quality than using a universal pattern?) In the experiments, the pattern was chosen to be learned with one particular noise level. Would learning in different noise levels produce drastically different patterns? If so, why one would choose one particular pattern to another? Would there be a trend in generated sensors across different noise levels that could be learned? A temperature factor is used when learning the initial sensor layer. What would happen if this factor \alpha_t is sensibly slower/faster than the chosen warming rate? Does the rate of increase of this factor condition the number of effective possible iterations during training? How is \gamma chosen? In the reconstruction network, what is the motivation for learning the color combinations and their gating values in two separate paths? What is the meaning of K, why is K chosen to be 24 in the experiments? Is the log-domain used to ensure positiveness of the color channels? In the second path, for learning the gating values \lambda, why the number of layers is chosen to be 128? Perhaps a naive question: why choosing nips over a vision/photography conference?

Confidence in this Review

2-Confident (read it all; understood it all reasonably well)


Reviewer 2

Summary

This paper proposed to jointly learn a color filter array for color image acquisition and the corresponding demosaicking algorithm using CNNs. The proposed approach makes intuitive sense and gives good results.

Qualitative Assessment

Typical cameras record RGB images using Bayer patterns, and reconstruct the RGB image via a demosaicking algorithm. This paper proposes replacing the Bayer pattern (or a hand-design different pattern as in [4]) and the corresponding demosaicking algorithm by patterns and demosaicking algorithms jointly learnt by a CNN. This makes sense, the method is sound, and the results obtained are good. My only concern is in the evaluation, where the demosaicking method of [26] is used, whereas better methods are clearly available (e.g., the LSSC method of Mairal et al.'09, and probably other methods since then). The authors should clarify this issue.

Confidence in this Review

3-Expert (read the paper in detail, know the area, quite certain of my opinion)


Reviewer 3

Summary

Author propose learning sensor design jointly with a neural network that allows carrying out inference on the sensor’s measurements. In particular, they focuse on the problem of finding the optimal color multiplexing pattern for a digital color camera. Authors learn this pattern by means of a NN for reconstructing full color images from the multiplexed measurements. For this, they make use of a soft-max operation with an increasing temperature parameter to model the non-differentiable color channel selection at each point, which allowed them to train the pattern. They finally demonstrate that their learned pattern enables good reconstructions.

Qualitative Assessment

Things to do to improve paper quality: 1) In sections 3 and 4, mode details need to done in order to reproduce the experiments by the interest reader.

Confidence in this Review

2-Confident (read it all; understood it all reasonably well)


Reviewer 4

Summary

This work introduces a jointly learning model for optical measurement and computational reconstruction. From machine learning perspective, the novelty of this work is limited.

Qualitative Assessment

Q1. The learning-based model has a drawback which is prone to has a bias with respect to the training data. It would better to test it on at least two dataset and have a discussion on the problem of generalization. Q2. How to learn discrete numbers, i.e., 0-1, instead of real values in the training is one of key challenges for optical measurement. It would be better to provide a further investigation of the influence of alpha on I(n) along the iterations. For example, plot alpha vs I(n) with respect to a certain color channel. Q3. Comparing to common CNNs training on ImageNet, iterations are much huger while the size of dataset is smaller, it would be better to discuss about why the proposed model needs so many iterations. (By default, Caffe runs total 450,000 iterations for Training AlexNet on 1.2 million images while the proposed model needs to run 1,500,000 iterations on 461 images.) Q4. Since alpha tends towards infinity, w may change rapidly. Can the proposed model converge to a stationary point, i.e., I(n)*? Q5. This work follows [4] in terms of experiment setting and [4] provides plots to shows tendency of the results. Why does this work not to use Table 2’s data to generate similar plots? It would be more intuitive.

Confidence in this Review

2-Confident (read it all; understood it all reasonably well)


Reviewer 5

Summary

This paper tackles the problem of reconstructing an RGB image from RAW data captured by a camera. Traditional approaches use a Bayer pattern to capture a single color intensity at each sensor location and a demosaicking algorithm to computationally reconstruct the colors at the pixel where they were not measured directly. This work uses a deep network to learn the optimal color sensor multiplexing array and a reconstruction algorithm to complement it. The network is trained by backpropagating the gradients from a pixel-level reconstruction loss. The multiplexing color pattern is encoded as a layer in the network, which selects one out of C channels at each sensor location using the softmax operation. The output is then passed on to a reconstruction network that reconstructs the full 3-channel RBG image. The reconstruction network has two paths, one to output K proposals for each channel and the other to learn the weights used to convert these K proposals to a single output using a linear combination. The experiments are performed on a standard dataset and the results clearly show the improvements in performance achieved by the proposed method over existing approaches.

Qualitative Assessment

This reviewer feels that the approach taken in this work has some degrees of novelty and that some of the ideas in the proposed method can improve the image capturing process. 1. Learning the multiplexing pattern by encoding it as a layer in a deep network is novel. This data-driven approach allows for exploiting the natural statistics that might be present in the image capturing process, but which might not be apparent to manually designed filters. 2. The work employs some clever tricks such as using the softmax layer with a temperature parameter to make the distribution peaky, which allows for selecting one of out of the C channels. This trick might be useful in other problems and applications that involves making a one out of K selection. Some suggestions for improvements: 1. The experimental results reported in the paper are mainly the PSNR values. However, since image quality is a highly subjective measure, it would be better to report judgements by human subjects. That would make the results stronger. 2. It might not be entirely easy to reproduce the results using the proposed method. Also, there is no mention of code and models being released. Do the authors have any plans to make the training code and model publicly available?

Confidence in this Review

2-Confident (read it all; understood it all reasonably well)


Reviewer 6

Summary

This paper presents a deep neural network model for simultaneously learning CFA patterns and a good reconstruction from sensor inputs in digital cameras. The paper shows promising results as compared to methods that utilise static CFA patterns, e.g. Bayer.

Qualitative Assessment

This paper is very well written with only a small number of grammatical errors. To the reviewers knowledge the idea is novel and can definitely have a practical impact. However, the novelty of the paper in relation to the NIPS audience does not exceed a poster-level presentation. The results section would make a better case if it could show a better example reconstruction then the one shown in figure 3. The differences in the one shown is hard to see electronically and impossible in a paper format. Since this paper is primarily directed towards one application it would be nice to see results on one or several larger datasets in cohesion with a more thorough runtime analysis of the different methods.

Confidence in this Review

2-Confident (read it all; understood it all reasonably well)